# Rule-Driven Forwarding for Resilient WSN Infrastructures [note 1]

**DOI:** 10.3390/s22228708

**Published:** 2022-11-11

**Authors:** Pawel Gburzynski, Ioanis Nikolaidis

**Affiliations:** Department of Computing Science, University of Alberta, Edmonton, AB T6G 2E8, Canada

**Keywords:** wireless sensor networks, multi-hop forwarding, security

## Abstract

We present a simple, robust, ad hoc forwarding scheme for small-footprint, low-cost wireless sensor networks (WSNs) and discuss some of its features from the viewpoint of resilience. The class of applications shaping the resilience requirements for our discussion covers temporary, and possibly disposable deployments of potentially massive populations of sensing nodes to assist in the management of emergency missions, including the management of the effects from natural or man-made disasters. Our goal is to make the network resistant to failures of any of its fragments, which may result from the network’s intended modus operandi or from intentional malicious attempts at crippling its function, while keeping the cost and footprint of the devices at the absolute minimum.

## 1. Introduction

By a wireless sensor network (WSN), we understand a distributed system interconnecting a number of sensing devices with the primary intention of conveying their readouts to one or more collection points [1]. The readouts may be creatively pre-processed by collaborating cohorts of nodes, sometimes resulting in virtual sensors [2]; some devices may also be equipped with actuators [3]. The adjective “ad hoc” refers to the self-organizing nature of the network [4] and the lack of a hardware infrastructure. The amount of data exchanged is typically small: a message consists of simple numerical sensor readouts, or actuator settings, amounting to a few bytes. The frequency of the readouts/control messages is also at most moderate, typically of order 1 Hz or less, but the network often has to react to “out of band” events [5] such as alarms. The typical raw transmission rate of low power wireless transceivers is of the order of a few tens of kbps. We consider devices that operate in sub-1 GHz bands, as they present an acceptable compromise between transmission power and range and include many of the frequencies used for tactical networks. For the proof of concept demonstration, we restrict our attention to sub-1 GHz ISM bands. Nevertheless, proposals for the (less often) used 2.4 GHz band [6] exist as well.

In many practical areas of WSN the ad hoc (especially multi-hop) paradigm has been recently subsumed (or usurped) by the Internet-of-Things (IoT) bandwagon [7]. In our continuing partisanship of *true* ad hoc networking [8], we argue that IoT, and standardizations thereof, neither covers nor replaces ad hoc. There seem to be at least some niches where unhindered multi-hop ad hoc networking appears attractive. For example, ecological monitoring [9] or preventive sensing in forests [10,11], may involve networks not intended to interface to the Internet at all, or interfaced at a single and expensive access point, e.g., a satellite link. Their energy constraints often call for low-power operation organized into frugal, duty-cycled episodes of activity and communication [12]. To focus the present paper, we consider potentially massive WSNs deployed in an “ad hoc” fashion for the purpose of assessing and managing an active tactical environment, in times of conflict or in peacetime, e.g., as a response to natural or man-made disasters. The primary design goal is resilience. The WSN may constitute a part of a larger deployment, e.g., including command and control or emergency command systems [13], but our focus is on the *sensor data* exchange.

Within this context, we present a simple and robust ad hoc forwarding scheme for wireless networks built of small-footprint devices. The basic idea, dubbed TARP (tiny ad hoc routing protocol), was sketched elsewhere [14] and since then has been applied in a number of practical WSN solutions [2,5]. In this paper, we give a detailed description of the scheme, significantly enhanced and better understood through the feedback from its realizations, and address the issue of its resilience. More specifically, the novel aspects include: (1) formal definitions of the rules with their precise implementation-level parameterization (Section 4.3, Section 4.4 and Section 4.5), (2) recent security add-ons (Section 5.2 and Section 5.3), and (3) an analysis of the resilience to faults and attacks (Section 5.4 and Section 6.2). The network’s observed resilience serves a dual purpose: maintaining connectivity in the face of damage sustained in the course of normal operation, and protecting the network against malicious attacks on its function.

## 2. Constraints and Tradeoffs

Two basic constrains underlie practical WSNs: the energy budget and the cost. If the network consists of hundreds or thousands of nodes whose individual actions are meant to be simple, one would naturally expect them to be small and inexpensive. Moore’s law is less relevant to the domain of WSN [15], so the notions of “small” and “inexpensive” have been stable in this area over decades despite the technological progress. When the devices are likely to be damaged (as part of normal operating conditions) the issue of their per-unit cost is even more relevant. By having small unit cost, the cost determinant becomes the scale of deployment.

Many practical application of WSNs are concerned about the energy budget. In a network of a more permanent nature, e.g., deployed within a facility for operational support [16], one can typically see nodes of two kinds: static *Pegs* providing a semi-infrastructure and (potentially) mobile *Tags* often attached to (movable) equipment and/or people [5]. While the Pegs can be often powered from outlets off other energy-rich platforms, the Tags must depend on their own sources of energy (batteries or energy harvesting [17]), which makes their budget considerably more constrained.

In our targeted environment, most nodes will be powered from batteries. Their operation can be intense (for a WSN), and the network is expected to last for at most a few days. To see the energy budget in a context, consider a typical, representative, industrial-grade device applicable for our purpose: CC1350 by Texas Instruments [18], at the unit cost of ca. $3, featuring an ARM-based microcontroller and a configurable radio (RF) module which can operate in a number of modes, including custom (sub-1 GHz) ISM, Bluetooth, and ZigBee. The device is powered from a 3 V source; its ballpark current drain in the different modes is listed in Table 1. The energy budget is clearly dominated by the RF module. For example, assuming the battery capacity of 1000 mAh, the device will last for about 5–6 days with the receiver on. On the other hand, in the normal idle state, when the microcontroller is just waiting for an event (to be triggered by a sensor), the same battery will (formally) last for over 100 years, i.e., practically forever. Most sensors (acceleration, temperature, light, humidity, carbon [di]oxide, and so on) draw between a few microamps and a few milliamps. Usually, the readouts requiring a larger current can be easily duty cycled [19], bringing the average current drain per sensor to the level of a few μA.

We adopt the view that the longevity of a constantly active node (4–7 days) is enough for assisting operations, especially given that over the period of those days, the presence of units in the same area means that additional nodes can be deployed. In addition to cost savings, such options can diversify the network’s application, e.g., by providing for its smooth transition from a monitoring mode (with longevity expressed in years) to an “incident/disaster mode”, with a scalable increase in the energy budget. This kind of multi-modal operation is natural, e.g., for networks monitoring forests against fires [10,20].

The power budget of nodes equipped with actuators is more difficult to estimate in a blanket manner; however, it is rather obvious that such nodes are going to be special. For example, a node controlling an access gate is likely to be powered from an outlet, possibly assisted with a battery backup. A field node equipped with an actuator may control a dispenser, a light (e.g., for signaling), or a motor (e.g., for mobility). Its longevity is thus likely to be limited by factors independent of the networking issues.

While the RF module is usually the most greedy participant in the power budget, the transmissions contribute relatively little, with most of the power drain being incurred by awaiting for reception. This is because, depending on the network-wide organization of the duty cycles, the time of a packet arrival for reception can only be predicted with some finite accuracy. In the extreme case, when the node does not know when to expect a packet, it must keep the receiver on, which causes a large continuous current drain. For illustration, based on Table 1, a 1 s activity of the listening receiver costs the equivalent of ca. 20 packet transmissions at 38,400 bps.

Another important tradeoff factor is the coverage density expressed as the number of nodes per unit area. One obvious criterion is the RF hop range which, for a typical device in the interesting class is of order 100 m. While longer (kilometer) range can be achieved in open space (under some circumstances) [21], it comes at the price of reducing the transmission rate and should be deemed extremal. Considering the necessarily casual placement of the nodes, precluding careful planing, antenna adjustment, etc., we should rather opt for a safe and adjustable coverage based on some default range (e.g., 50 m) adjusting it “on demand”, where the terrain geometry, or the problem nature, call for a special treatment. One should keep in mind that the effective transmission range, even in combination with the redundancy needed to maintain network connectivity in the face of possible damage, is not the only criterion of node density. The mesh formed by the network may need to be dense for reasons related to the nature of the measurements and/or other actions carried out by the nodes. It does help when the routing/forwarding protocol is resilient to imbalance in coverage, i.e., flexible enough to take advantage of any superfluous nodes in the neighborhood while also coping well with occasional bottlenecks. Studies regarding the density of WSN deployments and node placement in rural areas have been reported in [20,22] (for fire detection) and [23,24,25] (for flood prevention). Their relevance to our discussion is limited, as they are related to specific hardware and communication schemes.

## 3. Resilience Issues in WSN

Within our focus, the problems of resilience, seen also as security and reliability, receive a slightly different flavor than in the most general context of WSNs [26] or the IoT [27]. More generally, the (otherwise natural) tendency to generalize, globalize, and standardize problems and solutions does not fare well within the domain of “custom” WSNs intended to cater to well defined classes of applications. This is what makes the area of practical sensor networking different from IoT where the global approach to protocols, security, and interfaces makes sense [28]. A specific WSN, however, seldom needs a general-purpose network layer solving all the possible communication problems among arbitrary subsets of nodes in the network, because its traffic patterns are strongly shaped by the application. Even some large and general classes of WSNs pose communication problems considerably different from traditional networking [29]. The network does need a communication scheme facilitating its application-specific traffic patterns and, not to put too fine a point on it, nothing else [8]. In this spirit, we have argued for a *holistic* approach [30,31] to building entire applications of WSNs (so-called praxes). Our forwarding scheme can be viewed as an embodiment of this philosophy.

When seen from the widest angle, the security issues in WSNs are complex and diverse [32]. Serious tradeoffs come into play when one wants to address them comprehensively in a way independent of the application. For example, any scheme involving certificates, public key cryptography, distributed trust, or cloud access [27], poses considerably more serious demands on storage, computing power, and/or infrastructure connectivity than a scheme based on a symmetric cipher [33].

Fortunately, in many practical scenarios, one can get away with a reasonably simple approach to security without compromising the application. In our case, owing to the restricted applicability of the devices, limited duration of the critical period of the network operation, and the absence of direct interface to the Internet, the network does not have to secure itself against every threat imaginable in a more open system. In particular, the internal security of communication, understood as the secrecy of the messages exchanged by the nodes, can be ensured with a shared secret key (or a number of shared keys) preconfigured into the nodes before the deployment. This will also take care of node authentication: any node knowing the shared secret will be presumed legitimate. The ”hardening” of the platform by physical means is relegated to the use of tamper-proof components. In our *holistic* approach to ad hoc forwarding, this simple authentication scheme is basically the only one that makes practical sense. Note that some security issues transpiring in comprehensive networks for disaster management and recovery, e.g., non-repudiation in command systems [13], do not apply in our setup.

Following the introduction of the essence of our forwarding scheme in Section 4.2, we discuss (in Section 5) the cryptographic tools needed to implement basic security and authentication in our WSN focusing on denial of service (DoS) attacks. DoS attacks in WNSs have been discussed at length in the literature [13,15,34,35], and the following generic types of threats have been identified: jamming, exhaustion, spoofing, and re-playing. Some of these terms may cover a variety of techniques, depending on the degree of penetration by the attacker and the mechanics of the network’s routing/forwarding schemes [35]. For example, an attacker able to spoof a legitimate node may be able to fool the neighborhood by advertising misleading routing opportunities. In this paper, we ignore DoS threats based on penetrating (reverse engineering) a network node and, e.g., learning the encryption keys. While such threats are of course real, their materialization requires time and other means that are unlikely to be available to the attacker. Even for microcontroller designs identified as weak and tainted with demonstrable security lapses, the exploits require many days to be carried out [36]. This criterion of effort applies to all the security components in our WSN. For example, the 32-bit message authentication code proposed in Section 5.2 would be considered weak in a high bandwidth network where collecting $2^{31}$ packet samples is not necessarily an impossible feat. The same effort in our network, at 10 different packets per second (a rather heavy traffic when observed from a single spot), would take about 7 years.

## 4. TARP

TARP is a holistic scheme whose objective is to pass packets over multiple hops in a wireless network. In the literal interpretation of ad hoc and holisticism, the scheme does not belong to any particular *layer*; it is simply “all there is”. Or, to fit within the terminology of recent decades networking literature, it would be considered an extreme case of a cross-layer protocol.

### 4.1. The Forwarding Paradigm

The classical formulation of the forwarding problem is this: given a packet received by a node, tell the identity of the node’s neighbor to which the packet should be passed on its way towards the destination. This formulation is natural and works well where the neighborhoods are sharp and stable, as in most wired networks. It naturally lends itself to layering by isolating the problem of hop-to-hop forwarding from end-to-end communication. Solving the end-to-end communication problem amounts then to computing and maintaining routes [37].

Owing to the multitude of simultaneous and conflicting criteria and constraints (energy [38], scalability [39], quality of service [40], and mobility [41]), it is impossible to discuss in this paper even the most important representatives of the various techniques proposed in the academic literature (see [42] for an overview most pertinent to the context of WSNs). The two unifying features of all those schemes are: (1) routes understood as explicit paths in the network, and (2) data-link-layer encapsulation, i.e., a packet sent on every hop is data-link addressed to a specific neighbor of the forwarding node. Even if, in an acknowledgment of the fuzziness of paths in the unkempt environment of WSNs, some routing schemes attempt to provide multiple, alternative routes [43], the “route” concept (and the consequent data-link encapsulation) is always there.

TARP embodies a departure from the route view and is based on the following assumptions and observations:Wireless communication, especially in WSNs deployed in the field and/or admitting node mobility or node failures, is inherently flaky. The reliability of a multi-hop transmission over a non-trivial number of links is the product of the reliability of all the individual hops, so it scales poorly [44].In the massive collaborative system of a WSN, a node should be able to contribute to the joint effort of the *whole* network in proportion to its opportunities and abilities. Within the “path-routing” paradigm, a node can either contribute to a path *fully* or not at all.The network is low-bandwidth and deals with short data packets. Complicated schemes intended to isolate data-link transfers and prevent collisions among simultaneous transmissions in the neighborhood are not effective in this type of surrounding.The fact that a packet is “accidentally” overheard by a neighbor of the transmitting node, for which it is not “officially” intended, is not a flaw but a feature that should be embraced by the forwarding scheme.Redundancy in forwarding a packet, understood as involving more nodes in this task than absolutely necessary, is in fact *absolutely needed* to make sense of multi-hop communication in a WSN (see Section 5.4).

At the level of a single hop, a packet in TARP is always broadcast (re-cast) to all neighbors of the forwarding node. Every node picking up a packet tries to find a reason *not* to forward the packet. An uninformed node may fail to find one, so it will act overly altruistically, its retransmissions being excessive and unnecessary. This may happen when the node has just joined the network, has moved, or has run out of memory to store the requisite information. Contrast this with the behavior of a similar node operating in the strict fixed-path forwarding regime. Until the problem is detected, diagnosed, and rectified, such a node will be completely useless as a forwarder.

### 4.2. The Rule Chain

TARP is driven by a chain of rules illustrated in Figure 1. The rules execute in sequence, and the first one that succeeds breaks the processing resulting in the packet being abandoned. Their philosophy is to capitalize on the above assumptions and observations in the following way:The implicit forwarding paths implemented by the rules are fuzzy and intentionally redundant. Their reliability does not depend on any of their individual components. In fact, TARP employs no traditional path concept.The contribution of any single node to the joint effort of maintaining connectivity in the network is gradually proportional to its opportunities and resources available.The rules focus on forwarding useful data with no administrative burden. Any administrative duties are carried out implicitly by the data packets as they are being received (or perceived) by the nodes.Any node overhearing a packet can use the overheard information to its advantage improving its opportunities for contributing to the global goal of maintaining network connectivity.The amount of redundancy in the rules is controllable in a mostly automatic way. Additional redundancy kicks in automatically as it turns out to be useful.

A rule may carry out some operations on the packet before succeeding or failing. For example, if the packet appears to be destined to the current node, the rule may pass it to the application. This qualification may be based on arbitrary characteristics of the packet, e.g., it can be associative [45] rendering traditional node addresses unnecessary.

### 4.3. The Cleaning Rules

Suppose that all we need is a simple mechanism to disseminate information among all nodes in a WSN. We can insert into the chain a single rule that passes every incoming packet to the application and fails. This way, every packet overheard by the node will be *received* and also forwarded to all nodes in the neighborhood. To prevent infinite flooding, we introduce a field in the packet’s (formal) header, denoted $H_{c}$, storing the number of hops experienced by the packet, and impose a limit on that number. $H_{c}$ is set to 1 on the packet’s originating transmission and incremented by 1 on every retransmission. In contrast to the traditional approach, where such a counter (usually denoted by *TTL*, for time to live) is decremented on every hop towards zero, we prefer it to reflect the number of hops made so far. The rule may look similar to this:


**Rule**
*
**LHC**
*
(limit hop count):

*if (packet.$H_{c}<$ MAX_HOPS) {*

   *packet.$H_{c}$ ++;*
   *receive (packet);*
   *return FAIL;*

*} else {*

   *return SUCCEED;*

*}*


Suppose now that we would like to extend the scheme to implement an efficient exchange of packets between identifiable pairs of nodes. As the first step of bringing it closer to a practical solution, we introduce three more fields into the packet header: two node addresses (source and destination, labeled *S* and *D*, respectively) and the packet’s sequence number (labeled *Q*). Consecutive packets sent by the same source have their *Q* values incremented by 1 modulo the counter’s range. It makes sense to encapsulate packet reception into a separate rule:


**Rule**
*
**RCV**
*
(pass the packet to the application):

*if (packet.D == my_address || packet.D == 0) {*

   *receive (packet);*

*}*


*return packet.D == my_address ? SUCCEED: FAIL;*


It accounts for broadcast packets assuming that the special destination address 0 identifies packets that should be received by all nodes. Note that such packets are not removed after the reception: their fate will be decided by the remaining rules in the chain.

The subsequent rules reduce the redundancy of flooding. This one tries to make sure that a given packet is forwarded at most once:


**Rule**
*
**DD**
*
(duplicate discard):

*signature = concat (packet.S, packet.Q);*


*if (DD_cache.present (signature) {*

   *return SUCCEED;*

*} else {*

   *DD_cache.add (signature);*
   *return FAIL;*

*}*


The rule consults a cache of packet *signatures* composed by concatenating the packet’s source address *S* with its serial number *Q*. If the signature of a received packet is present in the cache, the rule succeeds causing the packet to be ignored. Otherwise, the new signature is added to the cache and the rule fails.

In a rule-based forwarding paradigm, the location of a specific rule in the chain may be flexible or it may be constrained by the locations of other (interdependent) rules. Note that *LHC* and *RCV* amount to the minimum sequence of rules in a workable network. Without *LHC*, the network would just flood indefinitely (chocking itself up), and without *RCV*, the nodes would not be able to receive. By putting *LHC* in front of *RCV* we assume that stray packets are an obvious nuisance and they should be discarded first. By the same token, it makes sense to insert the (more subtle) *DD* rule after *LHC* but before *RCV* to reduce the likelihood of bothering the application with multiple copies of the same packet. Hence, the recommended ordering of the three rules presented so far is: *LHC*, *DD*, *RCV*.

Entries in the *DD* cache age out after some time to account for the finite range of *Q*. They will also tend to be pushed out by fresh entries being added to the cache (according to the FIFO policy), because the amount of memory designated for the cache is limited. Note that when the node runs out of cache memory, the scheme will fail gracefully, by letting through some packet duplicates. The node will become more helpful than it should be, but the impact of this degradation on the quality of end-to-end communication need not be detrimental, and it can even be beneficial, if the cache parameters are set right. Note that, owing to the failure-prone nature of *DD*, the rule does not render *LHC* irrelevant: needing no local resources, the latter rule is always reliable. The reader should note that until this point, the benefits from TARP come from the comingling, at a local node level, operations that would have been performed at the data link and the network layer in traditional networks. Next, we show how the TARP rules can be used to balance, in a continuously dynamic manner, routing efficiency with reliability.

### 4.4. The *SPD* Rule

We now consider the operation of directing a packet towards its destination. The introduced TARP rule is dubbed *SPD* for Suboptimal Path Discard.

Imagine node *K* picking up a packet sent by node *S* and addressed to node *D*, as illustrated in Figure 2. According to Section 4.3, the packet identifies the source *S*, the destination *D*, and the number of hops $H_{c}$ taken by the packet so far. *K* sets $H_{SK}=H_{c}$ and stores the pair <$S,H_{SK}$> in a cache. The node has learned that it can be reached by *S* in $H_{SK}$ hops. Owing to the *DD* rule, operating before *SPD*, $H_{SK}$ tends to reflect the smallest number of hops via which *K* can be reached from *S* at the time. Similarly, when *K* sees a packet dispatched by node *D* it will store in the *SPD* cache the respective pair <$D,H_{DK}$>. Note that only the packet’s source matters for updating the *SPD* cache.

One more field in the packet header, $H_{b}$, is set by the packet’s originating node to the backward hop count, i.e., the last seen number of hops from the destination to the source. If the value is unknown (no relevant entry is available in the cache), the source will apply a large setting exceeding any legitimate number of hops, e.g., *MAX_HOPS* used by the *LHC* rule (Section 4.3).

Suppose that *K* wants to decide whether it should re-cast (forward) a packet on its way from *S* to *D*. The node calculates $H_{t}=H_{c}+H_{DK}$ and compares $H_{t}$ with $H_{b}$ extracted from the packet’s header. If $H_{t}$ is larger than $H_{b}$, the node has grounds to suspect that there is a better route for the packet than a path passing through *K*. This is because the estimated number of hops separating *S* and *D* is $H_{b}$, but when forwarded by *K* the packet is expected to make $H_{t}>H_{b}$ hops. If the information cached by the nodes were reliable and accurate, the rule might simply decide to succeed (thus dropping the packet) when $H_{t}>H_{b}$ and fail otherwise. This way packets would be forwarded along the shortest paths, although in a fuzzy manner, exploring multiple, alternative options. The redundancy of those alternatives can be adjusted with the *slack* parameter, denoted by *A*, interpreted as the acceptable excess of the path length over the estimated minimum. The rule succeeds if:(1)$$H_{t}>H_{b}+A$$
and fails otherwise. This admits multiple fuzzy paths up to *A* hops longer than the currently estimated minimum.

The rule includes a relaxation mechanism to prevent lockouts caused by stale data. The mechanism is controlled by a parameter, denoted by *C*, representing the number of successful applications of any given cache entry (resulting in dropping the packet) after which the rule should be gradually relaxed. To this end, every cache entry includes a counter incremented on every success of the rule triggered by that entry. The value of that counter is divided by *C*, and the result is added to the right-hand side of Inequality (Equation 1) together with the slack *A*. The effect is to locally bump the slack by 1 every *C* successes of the rule. When C=0, the relaxation mechanism is disabled. The complete rule looks similar to this:


**Rule**
*
**SPD**
*
(suboptimal path discard):

*SPD_cache.add (packet.S, packet.$H_{c}$);*


*ce = SPD_cache.get (packet.D);*


*if (ce != NULL) {*

   *R = (C == 0) ? 0: ce.SCount / C;*
   *if (packet.$H_{c}+ce.Hops>$ packet.$H_{b}+A+R$) {*
      *ce.SCount++;*
      *return SUCCEED;*
   *}*

*}*


*return FAIL;*


The counter (*SCount*) is initialized to zero when the cache entry is created or updated.

Note that *SPD* should follow *RCV* in the rule chain. This is because before being submitted to *SPD* the packet has been classified as a bona-fide original packet receivable by the node (based on *LHC* and *DD*), and the remaining issue is its retransmission.

The number of entries stored in the *SPD* cache will depend on the number of different sources in the packets overheard by the node. While this may be reminiscent of routing tables used by fixed-path forwarding schemes, such as AODV [46], one should note two contrasting features. First, the cache only stores one simple entry per (overheard) source (the size of this entry is eight bytes in the implementation) and does not have to deal with the node’s neighbors. Second, what is considerably more important, a node that runs out of memory for its *SPD* cache is not going to break the connectivity of TARP’s fuzzy paths. Similar to the *DD* rule, the node will then err on the side of overzealous forwarding, rather then refusing to forward, what would unavoidably happen in a fixed-path scheme.

Figure 3 illustrates the operation *SPD* in collaboration with *DD*. If A=0, then the exchange between *S* and *D* will be sustained through nodes $K_{1}$ and $K_{2}$. Although $K_{3}$ is overhearing the packets transmitted by *S* and retransmitted by $K_{1}$, it will find that the best path passing through it amounts to four hops, while the current shortest path length indicated in the $H_{b}$ field of those packets is three hops. If *A* is set to 1, the path leading through $K_{3}$, $K_{4}$, and $K_{5}$ will become active. In particular, *D* will be receiving packets both from $K_{2}$ and $K_{5}$, rejecting those arriving from $K_{5}$ as duplicates. If the shorter path fails (a packet that should arrive from $K_{2}$ does not materialize at *D*), its would-be duplicate reaching *D* from $K_{5}$ will not be rejected, so the secondary path will take over seamlessly.

Note that addressing only applies at the end-to-end (network) level; in particular, an internal node that never originates or absorbs traffic, needs no address. It can still receive broadcast packets, or overheard ones addressed to other nodes, if they fulfill its reception criteria. This makes the TARP idea somewhat reminiscent of pipelined forwarding schemes [47,48], intended for single-sink networks, where the forwarding decisions are based on the node’s *grade* related to its hop distance from the destination (sink). Another feature making those schemes similar to TARP is the amalgamation of the MAC layer into forwarding. Two important differences are: (1) the explicit notion of the number of hops separating a node from the destination used for partitioning the set of nodes into disjoint sets (interpreted as a static parameter); (2) the RTS-CTS isolation applied by those schemes at forwarding steps.

### 4.5. Enhancements

As the nodes overhear packets traveling in the neighborhood, they fill their caches, which helps them avoid unnecessary forwarding. The notable feature of the scheme is that lack of knowledge is not fatal, and the default behavior of an uninformed node is altruistic. One may raise three concerns against this approach:There is no isolation of the next-hop recipient, and every transmitted packet is essentially broadcast to all neighbors. This precludes collision avoidance schemes based on handshakes and may lead to broadcast storms [49], especially when the nodes are uninformed.It is not clear how much the flooding can be contained via heuristic rules that never attempt to construct precise, hop-by-hop routes. Is it possible at all to keep the redundancy at a level comparable to single-path forwarding?As a consequence of point 1, it seems that the only way to increase the reliability of forwarding is to involve more nodes (and network resources) into the process. With explicit forwarding to a specific next-hop-neighbor, the two nodes can resort to acknowledgments to improve the reliability of communication. This approach seems to be out of reach of the present scheme.

In the following paragraphs, we incrementally address the above concerns. First, to curtail the negative connotations of flooding, let us note that all forwarding protocols, including those that insist on rigid paths, have to deal with the lack of information at the nodes that otherwise might be able to help. Thus, at some stages, they all resort to flooding, if only to discover the neighborhoods and construct the explicit routes [50]. Put differently, flooding in the context of an uninformed node is no less natural to a scheme based on explicit paths than it is to TARP. In TARP, however, the node’s transition from uninformed to knowledgeable is never sharp and it triggers no fundamental mode change in the node’s behavior.

In answer to concern (1), it has been observed that RTS/CTS handshakes tend to be worthless and harmful when data packets are short [51,52,53]. In a mesh network, the incurred false blocking on exposed terminals is additionally pernicious [54]. WSNs built around TARP employ simple collision avoidance schemes based on LBT (listen before talk) and randomized backoff which work more than adequately for their niche applications [55].

With respect to concern (2), formally, the *SPD* rule can be tweaked to closely approach shortest-path forwarding. In particular, when A=0, packets tend to be forwarded in such a way that the number of hops traveled by them is the minimum feasible under the circumstances, where “feasible” means “based on the quickest, demonstrated way to deliver a packet from the opposite end.” Nonetheless, in some configurations of the mesh, this may incur multiple, simultaneous, alternative paths of the same length involving more nodes than (apparently) necessary. For illustration, in the scenario depicted in Figure 4, the paths $F_{1}$, $J_{1}$, $J_{2}$, $J_{3}$, $F_{2}$ and $F_{1}$, $K_{1}$, $K_{2}$, $K_{3}$, $F_{2}$ offer the same minimum number of hops between *S* and *D*, possibly including “cross-hops” such as $K_{1}\to J_{2}$ and so on. Then, in accordance with the *SPD* rule, all the nodes on the two paths will be forwarding packets between *S* and *D* (because the rule will always fail at each of them). As soon as the paths cross, the duplicates will be eliminated by *DD*, but one may object to the replication of labor along $J_{1}$, $J_{2}$, $J_{3}$ and $K_{1}$, $K_{2}$, $K_{3}$. While this redundancy lies beyond the scope of *SPD*, it can be controlled with a simple mechanism. For that, we add a single-bit flag into the packet header, labeled *O* (for optimal path). The flag is set by *SPD* whenever the rule fails on Inequality (Equation 1), i.e., based on the information available in the cache, *and* when it would also fail if *A* were equal zero (note that if *SPD* fails for A=0, then it would also fail for A>0). In plain words, it means that the rule believes the packet is being forwarded on a path with the minimum feasible number of hops.

Suppose two “parallel” nodes in Figure 4, say $J_{1}$ and $K_{1}$, decide to forward the same packet. This means that the *SPD* rule has failed in both cases, the setting of $H_{c}$ in both copies of the packet is identical, the *O* flag in both copies is set. The copies materialize at the two forwarders practically at the same time, the nodes queue their packets for transmission, and the LBT mechanism kicks in along with the randomized backoff. One of the two nodes, say $J_{1}$ wins, the other one, $K_{1}$, waits for the backoff timer with its copy of the packet queued for transmission. When $K_{1}$ overhears the copy transmitted by $J_{1}$, it will remove its scheduled copy from the transmit queue. The operation can be implemented as a rule preceding *DD* and looking similar to this:
**Rule*****SPP***(suppress parallel paths):
*if (packet.O) {*
   *pkt = find_in_xmit_queue (packet.S, packet.Q);*
   *if (pkt != NULL) {*
      *discard (pkt);*
      *return SUCCEED;*
   *}*
*}*
*return FAIL;*

The reason why the rule must precede *DD* is that the packet (based on its signature, see Section 4.3) has been already received by the node, so *DD* is likely to succeed on it. The matching modification in the *SPD* rule is easy to devise, so we skip it for brevity.

For *SPP* to work, the “parallel” nodes must be within mutual range. If they are not, the redundancy will be eliminated on the first pair of nodes where the paths “touch.” It is unlikely that the network spontaneously features two long and never touching paths, both offering the same minimum number of hops between $F_{1}$ and $F_{2}$ (although situations similar to this may arise during a jamming attack, see Section 6.2). Then, one should be careful not to overtrim the fuzzy paths, because some degree of redundancy is going to be useful, especially if the network is being designed for reliability and security. Considering that the most likely attack against the networks considered in this paper is DoS, intentional and controllable redundancy in forwarding will be of advantage. Where the demand for bandwidth is moderate, one should be willing to trade it for reliability, not only to combat malicious attacks, but simply to improve the quality of service offered to the application.

The local impact of *SPP* can be compared to the role of the traditional RTS-CTS handshake in point-to-point forwarding, whose objective is to eliminate damaging contention (collisions) in the neighborhood. While the RTS-CTS handshake is used to reserve the local medium for a single transmission to a specific neighbor, *SPP* works to eliminate excessive (redundant) retransmissions incurred by forwarding the same packet to multiple nodes in the same neighborhood. The effects of *SPP* are stronger because they translate into a reduction in the overall network bandwidth needed for successful forwarding.

For one more handy option, the relaxation feature of the *SPD* rule can be changed from local to global. As introduced in Section 4.4, the additional slack *R* resulting from the excessive success count of a cache entry augments the standard slack parameter *A*. This may force the rule to miss a “success” and retransmit the packet “out of band,” but the next forwarder will likely drop it because (based on $H_{c}$ and $H_{b}$) the packet will appear to have deviated from a feasible route. A stronger application of *R* is to add it to the packet’s backward hop count $H_{b}$. This has exactly the same effect for the local evaluation of the rule, but it also propagates the extra slack to the packet’s subsequent forwarders. The new variant is listed below:


**Rule**
*
**SPD**
*
(global relaxation option):

*SPD_cache.add (packet.S, packet.$H_{c}$);*


*ce = SPD_cache.get (packet.D);*


*if (ce != NULL) {*

   *packet.$H_{b}$ += (C == 0) ? 0: ce.SCount / C;*
   *if (packet.$H_{c}+ce.Hops>$ packet.$H_{b}+A$) {*
      *ce.SCount++;*
      *return SUCCEED;*
   *}*

*}*


*return FAIL;*


With respect to concern (3), while strict hop-by-hop acknowledgments (say, of the 4-way handshake variety [52]) are not possible in TARP, one can think of ways to improve the reliability of a fuzzy hop via multiple retransmissions driven by feedback from the neighborhood. Suppose that all the rules for a packet have failed and the node, call it *K*, decides to retransmit the packet. Having performed that, *K* will wait until it hears the same packet having been retransmitted by one of its neighbors with a higher value of $H_{c}$, which can be taken as an indication of progress. Until then, *K* will be retransmitting the packet at some intervals up to a maximum number of times. This option, dubbed *fuzzy ACKs*, has two parameters: the retransmission interval $T_{a}$ and the maximum number of (extra) retransmissions $R_{a}$ ($R_{a}=0$ means fuzzy ACKs disabled). One little hassle is that final destination of the packet must also retransmit it to notify the last-hop forwarder(s). It is enough to retransmit a dummy version of the packet only consisting of the header. The required information amounts to *S* and *Q* (the signature) plus $H_{c}$.

The *DD* rule can be strengthened (for all or selected packet types) to operate in the so-called shadow mode. With this option, the rule also succeeds when the *Q* counter in the processed packet is behind the value in the cache entry, but not further than by the shadow margin $\Delta_{s}$ (interpreted modulo the counter’s range). This assumes that a new packet from a given source renders old packets irrelevant, and makes sense for many event/alarm reports. The more aggressive operation of *DD* helps resolve packet storms, e.g., resulting from panicky multiple pushes of an alarm button [5].

## 5. Security and Resilience

Many advanced, cryptographic solutions to security problems in wireless communication introduce overheads that are unacceptable in WSNs. The problem was discussed in [56] and [33] where a lightweight set of tools was proposed and analyzed within the context of the resources available in tiny motes. While both those papers must be considered aged today, they still provide important guidelines for securing WSNs [57]. The basic constraints in terms of the energy/memory/processing power available to a wireless node are basically the same today as they were two decades ago. Even if the bar has been raised a bit, it always makes sense to try to minimize the amount of resources needed within the holistic framework of the WSN application [15]. This is one of the reasons why Moore’s law fails for WSNs.

### 5.1. Message Format and Types

The TARP rules outlined in Section 4 constitute the reference point for all communication scenarios in our WSN. The base implementation of TARP, devoid of security features, assumes the packet format shown in Figure 5.

The fields: *S*, *D*, *Q*, $H_{c}$, and $H_{b}$ have been covered in Section 4.3, Section 4.4 and Section 4.5, with the single-byte *F* field devoted to the message type (5 bits) plus three extra flags, including the *O* bit (Section 4.5). *L* is the length in bytes of the portion following *L* and including the *CRC* field guarding the packet’s integrity. The *NID* field acts as the network Id separating multiple networks operating in the same neighborhood on the same RF channel. This packet format was first used with the CC1100 radio module [58] where the maximum (raw) packet length (not including the *L* field) was 62 bytes. That imposed the 50-byte limit on the payload length. As the CC1100 module has been the preferred choice in a number of practical WSN systems built around TARP [5], that constraint has underlain the application design and has been considered hard. Note that it roughly agrees with the assumptions made in [33] where the maximum payload length was 29 bytes. The larger bound in our case can be interpreted as an advancement in technology, but it should not be enthusiastically embraced as a step forward. Most sensor reports are just a few bits; thus, most payloads are equally short, and it would be nice if the complete packets were not much longer. Shorter packets translate into less traffic, less noise in the neighborhood, lower chance for a reception error, smaller amount of buffer space, and so on. Any inflation of the packet header, if it can only be avoided with a reasonable effort, appears as an unnecessary overhead detrimental to the overall performance.

Without presuming too much about the application, we can try to identify certain generic types of messages (packets) that the network may want to pass around. The following list is inspired by our generic Tags & Pegs blueprint providing the basis for several real-life applications [2,5]. The network is governed from the master node acting as its interface to the outside world.

Master beacon. This type of message is issued periodically by the current master node to announce its identity and distribute network-wide parameters. The payload can vary from empty to potentially the maximum, with the expected length close to zero. It is one of the very few network-wide broadcast messages.Report. The message originates at any node and is addressed to the master. It can be *acknowledgeable* or not. In the former case, the issuing node expects an acknowledgment from the master.RPC. This is a request from one node to another node (a Remote Procedure Call, with the recipient possibly located several hops away from the sender. An RPC can be acknowledgeable or not.Ping. This message is addressed to the neighborhood and is not intended to be forwarded (as such) to any particular destination. A ping can be acknowledgeable or not.

Ping-type messages may connote hello packets exchanged in networks with explicit neighborhood/route discovery mechanisms, but the idea is quite different. A frugal Tag node operating on a tight energy budget may not participate in forwarding. Such a Tag will only turn on its RF module when it has something to say, and it will convey its message as a ping intended for any Peg in the Tag’s immediate neighborhood. The Pegs receiving the ping will transform it into a report forwarded to the proper destination node, i.e., usually to the master.

Pings can be acknowledged. Following the transmission of an acknowledgeable ping, the Tag will keep the receiver open for a short while (typically less than a second) expecting an acknowledgment from a Peg. Multiple Pegs may legitimately pick up the ping, but the Tag only cares about one ACK (from any of them). The Tag may repeat the ping a few times until its reception is acknowledged by a Peg. Not all pings need to be acknowledged. For example, in the network described in [5], unacknowledged pings are used for location tracking.

The ping-ACK mechanism is also used for passing messages, e.g., parameter change or actuation, to energy-frugal Tags. When the master wants to convey a message to such a Tag, it will notify the Pegs that have recently reported the Tag’s readings (triggered by pings). At the nearest opportunity when a notified Peg receives an acknowledgeable ping from the Tag, it will insert the message into the acknowledgment.

The *SPD* rule gives a preferential treatment to the traffic addressed to the master by locking master-related entries in the *SPD* cache. The propagation of a master beacon sets up entries in the direction *to* the master. This facilitates traffic to the master which in most networks accounts for over 90% of all traffic.

RPC-style communication between a pair of nodes, neither of them being the master, is used in local collaborative scenarios. A node initiating an RPC exchange may trigger superfluous forwarding, and possibly even flood the network (if the recipient has not been talking recently, e.g., sending reports to the master), but the response from the recipient will be assisted by the *SPD* entries collected on the first (request) message. This is not completely unlike route discovery in a network employing rigid-paths.

### 5.2. Confidentiality and Authentication

The natural reflex when network security becomes a concern is to encrypt traffic. This may provide confidentiality, which is not necessarily synonymous with security as demanded by the application. Confidentiality of sensor/actuator data may not be important at all. For example, the contents of communication between a proximity sensor and a nearby gate controller can be determined by looking at the gate, as can be the interaction between a motion sensor and an audiovisual alarm. While one can think of scenarios where the confidentially of such messages may become important, the primary concern is usually authenticity, integrity, and trust.

If we were to adopt a confidentiality-first stance, we could trivially, encrypt the entire contents of a packet (including headers). Then, the very first rule in TARP would involve the recognition of the packet as being encrypted with one of a handful of known keys and dropping the packet on failure. Instead, following the spirit of [33], we put stress on authentication and provide confidentiality as an option. The former is implemented via a message authentication code (*MAC*) replacing the *CRC* as a cryptographically secure enforcement of packet integrity. As the most natural ways to implement *MAC* involve block ciphers [59], the optional confidentiality will come as a cheap add-on to the authentication scheme.

The layer-less nature of TARP precludes certain types of cryptographic paradigms and solutions. For example, the key distribution scheme proposed in [56] assumes that the “base stations” in the network are trusted and their resource constraints are relaxed compared to those of the sensor nodes. Additionally, a base station is assumed to be authoritatively responsible for the nodes falling under its jurisdiction, implying permanent or semi-permanent bonds, which is exactly what TARP abhors. While some networks driven by TARP may be naturally built around a semi-infrastructure of Pegs powered from inexhaustible sources [5], networks deployed in the field cannot afford this luxury, even though conceptual equivalents of base stations (sensor-less Pegs) can be identified in them. The TinySec suite introduced in [33] is frugal about the prerequisites and concerned about resources, but it still refers to the “link-layer.” Although one can agreeably try to equivalence the forwarding paradigm of TARP with the traditional data-link layer, it is clearly not the same thing. As we noticed at the end of Section 4.4, a forwarding node need not even be equipped with an address. Thus, for example, the data-link-layer-based security solutions mentioned in [60,61] cannot be applied in our network. The authors of [33] do recognize that the link-layer in WSNs is a potentially tricky issue, so they rightfully argue for a single-key, network-wide encryption/authentication scheme for hop-by-hop communication. We strongly agree, especially that TARP admits no other option.

The block cipher used as the cryptographic basis for our scheme is AES [62]. A technological advance with respect to [33] is that efficient software implementations of AES are now available [63] and, more notably, contemporary commodity microcontrollers come with hardware AES. The nominal bandwidth of the AES engine available on CC1350 (our microcontroller of choice for the updated, secure version of TARP) is 118 Mbps [18]. A CBC encryption of a 32-byte packet (using a pre-scheduled key), including all overheads, takes less than 10 μs. In confrontation with the amount of time needed to handle that packet in a node and transmit it over RF (at least a millisecond), the energy expenditure on encryption is negligible.

To enforce semantic security, we have to make it unlikely that the same message encrypted twice will produce the same ciphertext. For that, we need an initialization vector (IV) that will tend to be different for different packets. Owing to the fact that packets transmitted in our network are numbered (the *Q* field), we can naturally take advantage of those numbers to increase the IV space, in a way similar to [33]. The original range of *Q* (Figure 5) is one byte which is not enough for the purpose. The analysis carried out in [33] suggests a 16-bit counter combined with other fields from the packet header. We opt for a different approach which will bring in an additional feature, explicitly omitted in [33], namely playback resistance. As the master beacons constitute an essential component of the communication scheme, we shall use them to implement a fuzzy global clock. By including a fuzzy time stamp in the packet header, we will provide an additional differentiating value for the IV and render playback attacks ineffective.

The modified packet format is shown in Figure 6. The essential differences with respect to Figure 5 are: the new field *T* storing the time stamp, the replacement of *CRC* with *MAC*, and the removal of *NID*. The entire packet header, including the length field, is used as the IV for the CBC mode of AES. Suppose the message needs to be encrypted. The 11 bytes of header are padded with 5 zeros to form a 16-byte block for AES and then encrypted to establish the IV to be subsequently applied for encrypting the payload. The reason why the IV is encrypted before its application is to eliminate counter leaks [64].

The new packet format is intended for CC1350 where the maximum packet length is formally 256 bytes. We have decided to explicitly avoid packets with payloads larger than in the original version (Figure 5), partly for application compatibility and partly because long packets are incompatible with the spirit of our WSN (Section 5.1). For the same reason, we do not want to encrypt all payloads blindly, cheap as it is with hardware AES. The minimum length of a payload that can be encrypted is 16 bytes (the block size of AES). A partial block beyond the minimum size is OK: ciphertext stealing can be applied in such a case [65], so as long as the payload is at least 16 bytes long, the encryption will not increase its size; however, many messages are shorter than 16 bytes, and they would have to be inflated for encryption, which discourages its indiscriminate application. A single-bit flag in the *F* field can be used to indicate whether the payload is encrypted; the application may also assume that messages of some types are always encrypted while others are not.

Note that the header is never encrypted, because it acts as the IV for encrypting the payload and must be available before decryption. This leaks information about traffic patterns in the network, even when the payloads are encrypted, but we have agreed that this is much less important than message authentication. For *MAC* calculation [59], the IV (padded with zeros to 16 bytes) is concatenated with the payload (padded with zeros to a multiple of block size) and encrypted. The first four bytes of the resulting ciphertext are then inserted into the packet’s *MAC* field. The arguments put forward in [33] in favor of a 4-byte *MAC* stand also in our case. As the *MAC* is cryptographically secure and twice as large as the previous *CRC*, the *NID* field can be eliminated. Different networks will be using different keys, so they will be separated by the incompatibility of their *MAC* values, even if they happen to operate in the same area and on the same channel.

In extremely sensitive scenarios, specific end points exchanging packets (e.g., *RPC*) can use separate keys (different from the network-wide key used for forwarding) to encrypt their payloads. This may preclude the application of tricky rules at the forwarders, which are not going to see the payloads, but is an easy solution for the circumstances where extra security is needed.

### 5.3. Replay Protection

An intruder may store copies of overheard messages and transmit them later into the network, thus conducting replay attacks [66]. If the intruder knows what the messages mean, she can try to force the system to carry out specific malicious actions. Otherwise, she may just try to confuse the network.

In [33], the problem of identifying replayed packets was “explicitly omitted” and delegated to the application because the proposed scheme was generic and the authors did not want to impose solutions (inflating the packet header in a rigid way) that the application might not be comfortable with. We, in contrast, can take advantage of two facts specific to TARP and to the way our applications tend to be organized [2]:All packets passed by TARP include a *Q* field acting as a small-range, wrap-around counter for packets sent by a given source.The concept of periodic master beacons is an integral feature of all TARP networks where centralized data collection is of merit.

Using the *Q* field alone as a measure against replay is problematic, because it is only effective within a small window. Even if its range were extended, it would not provide a foolproof safeguard against replay attacks, because the size of the *DD* cache is limited. Without other means, the attacker could replay many packets coming from different areas of the network, thus overflowing the cache, which would cause some entries to be removed. Then, any packet purporting to originate at any source absent in the cache would be interpreted as a bona-fide new packet.

Instead, we propose to mark the packets with 16-bit, second-grained time stamps issued by their originating nodes. Those time stamps are crudely synchronized by the master beacons which carry (within their payload) the 32-bit wall clock of the master. The 16-bit range of the *T* field in the packet header provides for ca. 18 h wraparound time. If this is a problem, the application can apply some (possibly crude) criteria to identify and discard (extremely) old packets [66]. An obvious and cheap remedy that immediately comes to mind is to include the upper 16 bits of the full time stamp in the packet payload.

A node receiving a packet whose time stamp is a few seconds out of line compared to the node’s own clock will ignore the packet before consulting any caches, so the packet will cause no stress on the node’s resources. The same node will always trust the master beacon whose protection against replays is guaranteed by the always-increasing 32-bit second counter.

An intruder trying to overcome this mechanism would have to replay fresh packets collected at most a few seconds ago; however, by the very nature of TARP, this cannot be harmful. Any duplicates will be promptly rejected based on the current contents of the caches, and if, by accident, the original packet did not make it to the node, the intruder will be actually helping the network (!), in the same way as a legitimate uninformed node would.

The combined net increase in the packet frame size over the unprotected variant of TARP is 2 bytes. They enter into the *MAC* field to act as a cryptographically strong replacement for *CRC.* A *MAC* failure in a received packet can mean one of three things:A reception error, i.e., some bits in the packet have been corrupted.The packet belongs to a different network operating in the same neighborhood on the same channel (the *MAC* doubles as the *NID* from the unsecured variant of TARP).An intrusion attempt.

Option 2 can be excluded in deployments. If there is a need for multiple networks operating in the same neighborhood, they can use different channels which (in the sub-1 GHz ISM band used by CC1100/CC1350) can provide for at least 50 different networks coexisting with minimal interference. Thus, nodes may detect a suspiciously large frequency of *MAC* failures, especially for packets with strong RSSI (Received Signal Strength Indication) and report them as suspected attacks.

### 5.4. DoS Attacks

Owing to the nature of TARP, some types of DoS attacks, normally discussed in the context of WSN, are not possible in our setup. Among those are fake RTS/CTS packets [67], hello floods, ACK spoofing, dummy connection requests, and, generally, any attacks trying to fool the (non-existent) procedures of the medium access control or data-link layers involved in routing and forwarding [35]. Assuming the security of network-wide encryption and authentication (Section 5.2), and resistance to replay attacks (Section 5.3), the only practically conceivable attack type (to which all wireless networks are exposed) is straightforward jamming [68] where the attacker does not need to understand, penetrate, or break any mechanisms within the network. Some realizations of the physical layer (spread spectrum, frequency hoping, redundant encoding, filtering) may render jamming more difficult [69]. TARP is largely transparent to such schemes, so they are applicable in our case (e.g., redundant encoding can be selected as an option in the CC1100/CC1350 driver [18,58]). One exception is that some frequency hoping techniques assume data-link level synchronization between the transmitter and the receiver, which TARP cannot afford.

Intelligent jamming attacks may include periodic jamming, scheduled at specific time instants and intended to maximize the damage while minimizing the energy expenditure of the attacker, or attempts at energy-starving some nodes. Jamming the master node could be a particularly attractive option to the intruder. The likelihood of success on this front will be reduced when the master is placed in an area that cannot be easily penetrated by an attacker. This makes sense because the master node is usually directly interfaced to an external platform/computer, if only for the acquisition of accurate timestamps, e.g., from GPS. Jamming a group of nodes when they are about to receive a master beacon is another option. To make this more difficult, the beacon interval can be randomized.

Starvation attacks, as such, only make sense in a wake-on-radio (WOR) duty cycled network where nodes wake up in response to radio signals [70]. While this kind of operation is possible in TARP networks, we do not envision it for the class of applications considered in this paper. It is more likely (especially in the no-alarm, monitoring mode) that the application will deploy semi-synchronized duty cycling orchestrated by the master, but then the master beacons may advertise the schedule in an encrypted message, or follow a pre-agreed, cryptographically secure pattern derivable from the time stamps and the network-wide key. Then, the intruder will have to listen to the channel continuously (or almost continuously) to sense a burst of activity in the network worthy of jamming.

From the viewpoint of the assumed application context, a jamming attack is symptomatically equivalent to the disappearance of a network segment, which is indistinguishable from the same happening (at least in some application) as part of pre-planned operation. Thus, the network’s resilience to jamming attacks can be immediately re-interpreted as a measure of its endurance in the face of justifiable damage.

Consider a jamming attack in the middle of the network against one of the (routing) Pegs. An attack of this sort basically disables a node or a subset of nodes in the neighborhood. In a network applying strict routes, especially one employing a routing hierarchy [71], disabling a small subset of nodes (e.g., clusterheads) may cause a serious disruption calling for complicated diagnostic mechanisms (attack discovery) and mending procedures (circumnavigating the disabled region), which actions take resources and time [34,71,72]. As demonstrated in the next section, TARP resolves most of such problems automatically and seamlessly, in the course of its normal operation.

## 6. Experiments

We illustrate the behavior of TARP in a generic WSN application dubbed Netting [73], intended for testing and demonstrations, run in an emulator. The emulator, called VUEE (Virtual Underlay Execution Engine), is a critical component of an elaborate platform for developing and testing WSN applications [2,30]. It executes the actual application code, i.e., the same program that would be flashed into real-life devices, within a virtual environment including a detailed RF channel model [74,75]. The experimental network consists of 1024 nodes placed on a planar grid with the separation of 40 m in either direction. The RF module can be assumed to be equivalent to CC1100/CC1350 operating in custom mode within the 916 MHz ISM band at the nominal transmission rate of 38,400 bps.

The propagation conditions correspond to an open, planar terrain with the nodes deployed about 1 m above the ground. The effective communication range is about 100 m (see Table 2). We assume that the primary purpose of the network is data collection with most traffic being addressed to the master node. The master (node 1) is placed in the left-upper corner of the grid; this position maximizes its average separation from a regular node.

### 6.1. Healthy Conditions

We start with a stable, healthy network driven by the sequence of rules introduced in Section 4.3, Section 4.4 and Section 4.5 with fuzzy ACKs switched off. The slack parameter *A* is set to 1, and the relaxation count *C* is set to zero, i.e., the relaxation option of *SPD* is turned off (Section 4.4). The report messages forwarded to the master consist of 16 payload bytes + 15 bytes of header and trailer (Figure 6).

Figure 7 shows two paths taken by two report packets consecutively dispatched by node 1024 (the one in the right-lower corner) to the master. This kind of orderly forwarding becomes possible after every node has received a single copy of the beacon packet broadcast by the master, i.e., the minimum possible amount of information making the node informed at all under any conceivable forwarding scheme. The packet sets a single entry in the *SPD* cache of the node.

The highlighted nodes are those that have retransmitted the packet. The two paths have been obtained within the same inter-beacon interval, i.e., they correspond to exactly the same contents of the *SPD* caches at all the nodes along the way, yet they exhibit considerable diversity resulting from the stochastic factors affecting the rules. Owing to the randomized LBT delays, different nodes forwarding the same packet may go first on different occasions. The *SPP* rule will cancel some of the scheduled transmissions before they materialize, and the *DD* rule will trim out late packets being received as duplicates. A long hop may sometimes succeed “by accident” and be tacitly taken advantage of without jumping to premature conclusions. The powerful simplicity of *DD* and *SPD* kicks in in such circumstances, aggressively exploiting lucky flukes, while keeping the more reliable forwarders ready to seamlessly help when the luck fails.

The *SPD* rule admits 1-hop detours (A=1) with respect to the minimum feasible number of hops which, in a grid network, similar to the one in Figure 7, offers a significant number of alternatives. Most of those alternatives are trimmed down by *DD* and *SPP*. The redundancy is partially *actual*, and we can see it in the paths (the number of nodes involved in passing the packet from the source to the destination is roughly twice as large as the number of hops taken by the packet), and partially *potential* (some transmissions have been preempted, so we do not see all the standby helpers). The actual redundancy may be easy to see (and perhaps object to), but its positive impact is difficult to overpraise. The average number of hops experienced by a packet traveling from node 1024 to the master is 21. The absolute minimum is probably around 11 hops (as suggested by the few sparse lucky spots in the paths), but betting on the odds would not be a good idea.

Table 2 shows the observed packet delivery fraction over a single hop measured against a random, steady, light background traffic amounting to one report message per second materializing at a random node to be delivered to the master. The distances correspond to eight feasible hop scenarios: next node (horizontally or vertically), one node across, two nodes sideways, two nodes sideways and one up/down, two nodes across the diagonal, three nodes sideways, three nodes sideways + one up/down. The packet loss rate across the longest hop is about 35%, so the probability of success over 10 such hops is practically zero. In TARP, where the paths are opportunistically, automatically, and statistically constructed from a large selection of potentially available hops, we are likely to see a few lucky hops in every actual path of a nontrivial length. However, no extraordinary luck is required for the entire path to materialize because the standby (less demanding) hops are there to take over without being asked. The PDF across the full diagonal of the grid is ca. 95%.

The unkempt character of paths in TARP is precisely what makes their redundancy constructive. Note that a rigid-route scheme utilizing data-link acknowledgments would exhibit a numerically similar redundancy interpreted as the number of extra transmissions (data-link retransmissions) needed to deliver a packet. A fixed route consisting of 21 steps may be built, e.g., of 10 long hops (112.8 m) and 11 short ones (56.4 m). Using the numbers from Table 2 and solving a simple Markov chain one can calculate the expected number of retransmissions needed to deliver the packet to the other end. Whichever reasonable way the path is split into 21 hops, that number exceeds the actual number of hops by the factor of 1.2–2.3, which matches or exceeds the overhead seen in Figure 7. In a fixed-route approach, the overhead is inflexible and stuck to the route. When the route fails for reasons beyond a casual packet loss (e.g., a portion of the network becomes disabled), the redundancy will not help until the route is rebuilt. In TARP, the redundancy is spread over the neighborhood, so it can kick in automatically when the previous best route becomes unavailable, for whatever reason.

It should be mentioned, especially in the context of jamming attacks, that the contents the *SPD* cache entries pointing towards the master are also randomized to some extent, and may differ slightly on the different passes of the beacon. This is because local congestion scenarios caused by the broadcast nature of the beacon packet cause statistical variations in the path length traveled by the beacon, as perceived by the same node on different occasions. This in fact *reduces* the severity of the jamming threat, because the impact of jamming at a particular location cannot be predicted too precisely (even if the population of disabled nodes is known exactly), so the attacker cannot know how successful the attack is going to be.

### 6.2. Network under Attack

The relevance of the *potential* redundancy comes into play when we emulate a DoS attack by disabling some nodes in the middle of the network. In Figure 8a, the grayed out group of nodes in the center have been switched off. No new master beacon has been issued (yet) following that mishap, so the forwarders have to cope with the problem without feedback! This is worth repeating: no detection, no recovery procedure has been carried out, and the nodes act on the same cached information as before. The path appears a bit more redundant than the previous one, but the packets are still delivered at the PDF of ca. 80%. Even when we introduce five more holes, as in Figure 8b, about 70% of the packets still make it to the master.

We are only forced to concede defeat under the rather massive attack visualized in Figure 9. While some packets manage to find their way around the nine holes (Figure 9a), less than 30% of them are delivered to the master (Figure 9b shows a failed path). The situation is remedied after the next beacon dispatched by the master triggers an update of the *SPD* caches at the operational nodes. Notably, the paths exhibited under the new conditions appear less redundant (Figure 10), which is not surprising: the network now offers fewer forwarding alternatives than before. Nonetheless, the delivery rate from node 1024 to the master is around 70%.

The role of the beacon can be compared to a recovery mechanism in networks employing rigid routing/forwarding schemes. A passage of the beacon following a mishap in the network will bring back the connectivity needed for report forwarding (by the nodes that are still in the game), as long as the network remains formally connected. However, as the beacon is a broadcast message flooding the entire network, one would prefer to issue it as sparsely as possible, e.g., at intervals expressed in minutes, especially that it is not much needed when the network stays healthy. The fact that the arrival of a new beacon is not required in many serious cases of disruption, and no explicit recovery procedure is needed, is advantageous and particularly helpful when facing dynamic jamming scenarios, e.g., when the attacker is moving around with a portable jamming device hoping to disable a significant section of the network before an action is taken, or, even worse, confusing that action and stalling its successful completion.

Several ways to adjust the parameters of TARP in preparation for an attack can be considered. Somewhat counterintuitively, fuzzy acknowledgments (Section 4.5), intended to improve the reliability of a single hop, are not very helpful. The intuition from strict forwarding is deceiving because TARP capitalizes on stochastic assistance by whichever nodes are lucky enough to pick the packet, instead of trying to account for every “solid” hop. In Figure 11a, we see stubborn insistence on a straight path towards the destination, and, sure enough, the path hits the large hole in the middle and stops there. The broken path is wide, redundant, and dense (A=1) because the mechanism of fuzzy ACKs machetes along the best hints supplied by the *SPD* rule. The multiple retransmissions along the “best feasible” path preempt most of the (potentially helpful) diversions. The network scrambles enough luck to sprout a narrow path circumventing the two holes, but, in comparison with the blocked “parade,” it has the appearance of a fluke.

A fresh beacon does help, as seen in Figure 11b. The forwarding is excessively redundant because there are multiple equally good paths around the holes, they do not touch, and the fuzzy ACK mechanism insists on steamrolling through them all. Generally, the option is unattractive as a blanket forwarding strategy. It has been successfully applied in irregular networks to assist with bottlenecks in selected subsets of nodes. When the connectivity between two sizable segments of the network can only be sustained via a narrow bridge of nodes, it makes sense to let those nodes employ fuzzy ACKs to improve the reliability of communication across the bridge.

Another idea potentially helpful under DoS attacks would be to increase the slack parameter *A*. In a relatively dense and regular network, where a given number of hops between a pair of communicating nodes can be achieved in many alternative ways, this tends to quickly increase the actual redundancy above the comfort zone. While statistically the *SPP* rule does a good job trimming down retransmissions along the multiple “best feasible” paths in adjacent regions, it does not help when the paths diverge into disjoint neighborhoods, which is likely to happen when they are inflated by large *A*. A better solution is to resort to the relaxation option of *SPD*. With the *local* variant of the option (Section 4.4), the slack applied by the rule is increased by 1 every *C* successes. This may trigger occasional redundant retransmissions by sufficiently many nodes in the neighborhood of a failing “optimal” forwarder to bring help. When the optimal forwarder succeeds, its transmission stands a good chance of preempting the assistance of the nearby helper, thus preventing the redundancy from becoming actual and spreading too far into the network. The impact of this feature is illustrated in Figure 12 showing two paths obtained under C=2 and A=0. The nonzero *C* causes a noticeable increase in actual redundancy (Figure 12a) with respect to A=1, C=0 (Figure 7) amounting to ca. 35%. This redundancy pays back when the network is attacked (Figure 12b). Even with the nine holes, the network delivers packets at the rate of ca. 75% with no fresh beacon from the master.

With the relaxation option on, the redundancy will tend to grow with the consecutive report messages dispatched towards the master, because more and more nodes will see the packets as they are being forwarded more and more generously by the ones preceding them on the path. However, the growth will be slow, because every subsequent potential forwarder will have to accumulate a gradually larger value of the counter, to account for the increase in the packet’s path length with respect to $H_{b}$. This is only relevant until a new beacon from the master updates the *SPD* cache entries at all those nodes resetting their counters to zero.

If the local variant of the relaxation option is insufficient, the global variant (Section 4.5) brings more help. Figure 13 depicts an RPC scenario involving nodes 364 and 661. The nodes have exchanged a few packets (both ways), and Figure 13a shows a plausible path from node 661 to 364 obtained under A=0 and C=2 using the local variant of the relaxation option. The path appears frugal, and the scheme does not seem to cause too much overhead under normal conditions. Then, a disaster strikes: the area is jammed in such a way that nodes 364 and 661 fall just on the boundary of the large circle of disabled nodes in Figure 13b. The relaxation option can still theoretically work to re-establish connectivity, but it takes about 40 futile attempts by node 661 to force a path to node 364 around the hole. This is because the propagation of local relaxation is extremely slow. The option is intended to remedy *local* connectivity problems, and the one at hand is not that local: a whole large set of forwarders between the communicating nodes has suddenly disappeared. The recovery is a bit faster (13 alternating attempts) when carried out from both ends (which is probably how it would be handled in real life).

Note that an RPC scenario is not helped by a master beacon, unless at least one of the two nodes decides to talk to the master following the beacon reception. To immediately reset the *SPD* entries related to the local session, one of the two nodes would have to issue a broadcast message, akin to the master beacon, which would propagate everywhere ignoring the old contents from the *SPD* cache. The response of the second node would then proceed according to the new information, resetting the *SPD* entries for the next message of the first node. While this may be considered as a measure of true desperation, a preferred solution is to apply a less drastic fix, one that would not put stress on the entire network, especially if it is already stressed by a jamming attack.

Figure 13c shows how the same problem is resolved by the global variant of the relaxation option with the same value of C=2. The undisturbed path appears slightly more redundant than before, but the disastrous mishap that materializes in Figure 13d does not break the communication. Note that the successful passage of the message from node 661 will update the cache entries in the reachable nodes, so they will immediately offer a decent route to a packet going in the opposite direction. That packet, in turn, will clear the way for node 661, so its next packet will not have to depend on the generosity of the relaxation option.

## 7. Conclusions and Final Remarks

We have described a fuzzy forwarding/routing scheme for ad hoc WSNs from the viewpoint of security in applications typical of tactical missions. The scheme, dubbed TARP, makes minimalistic assumptions about the resources available at a node, and smoothly scales to their availability and also to the node’s (intermittent) opportunities to assist other nodes in conveying their packets to destinations. The security tools added to the original (insecure) version of TARP incur very low additional overhead and, as we argue, are adequate for applications in detecting, diagnosing, and fighting disasters. In addition to providing a natural niche for truly ad hoc WSNs, such applications define specific safety constraints to which our proposed scheme seems to respond well.

TARP features controllable redundancy which allows it to gracefully and seamlessly respond to jamming attacks while being impervious to other types of threats, intentional or accidental, that the network may face in the course of fulfilling its duties. The efficacy of TARP in responding to jamming attacks is a side-effect of its way of harnessing the unreliability of wireless channels to implement effective and reliable multi-hop communication.

The paper does not discuss general performance issues related to TARP, focusing on the aspects related to security and resilience. A meaningful general quantitative performance comparison to other schemes, would have to account for many parameters including the distribution of nodes, channel characteristics, application demands (types of nodes, messages and their required delivery rates), and the parameters of TARP (which can be tweaked in several ways to tune the scheme to those demands), as well as those of the other schemes being compared. Additionally, such a comparison would still be open to criticism regarding its fairness due to factors such as TARP’s lack of separation between “data” and “control” messages, or TARP’s flexible approach to reliability, which are not necessarily present or acceptable in other schemes. Consequently, none of our previous published work related to TARP [2,5,14] compares its performance with that of other forwarding/routing schemes.

The discussion in Section 6.1 provides powerful arguments in favor of TARP in comparison to the wider family of forwarding schemes that depend on point-to-point forwarding. A possible scenario where TARP could be deficient to such schemes is one where traffic patterns impose hard delivery-time constraints (where the deterministic nature of fixed paths would become advantageous), or/and involve long data packets (where the data-link handshakes on the point-to-point hops would win over the simple LBT mechanism of TARP). Neither of these scenarios seem to be prevailing examples for typical WSN applications. TARP exemplifies a true adherent to the *cross-layer* networking paradigm. The real-life application of TARP presented in [5] provides a location-tracking functionality and is a revealing example that illustrates how the advantages of our scheme are fully realized via its holistic integration with the application.

While one can argue that testing the operation of our scheme against a network deployed on a regular grid (Section 6) is restrictive, we should keep in mind that the *internal* operation of the network’s transmission/forwarding scheme is based on inherently stochastic and unorchestrated reactions of the nodes whose timing and consequent serial configurations tend to depend on many essentially random factors. On top of the directly randomized LBT delays, many other incidents, such as a stochastic failure of a node to receive a weak signal, a fluke to successfully receive one, an *SPP* race, etc., may result in drastically different configurations of events down the road, each coming with its own source of subsequent randomization. This behavior (witnessed in Section 6.1 and Section 6.2) is in contrast to most rigid (fixed-path) schemes where the accusation of the regular network deployment would carry more weight. In our case, stacking two different randomizations on top of each other would increase the parameter space while adding no finesse to the model. Further, the regular grid deployment makes it easier to judge the overhead of TARP against fixed-path forwarding in Section 6.1.

The maximum technically achievable reception rate of the master node in the network introduced in Section 6 is about 60 packets per seconds. This figure results from the combination of the raw bit rate (38,400 bps), packet encapsulation, and the processing, and is practically independent of TARP. Needless to say, a nontrivial network, consisting of a hundred or more Pegs, cannot expect this rate to be uniformly shared by all nodes at all times, as it would cause a considerable congestion in the neighborhood of the master node translating into a poor PDF for all traffic; however, in some situations, the rate is achievable momentarily. For example, one function of the network presented in [5] was on-demand, alarm-triggered location tracking where a tracked Tag would emit a series of short unacknowledgeable pings (Section 5.1) to be picked up by whatever Pegs were available in its neighborhood and forwarded (as reports) to the master. In some circumstances that would cause intermittent storms where the master would in fact receive 50–60 packets per second representing a portion of those reports that could make it through the network. Even though the delivery fraction of the Tag pings and the Peg reports was low (sometimes below 10%), the application was prepared for that and could sensibly act on whatever information was received to carry out its task. The same application also required reliable reports from Pegs, which were acknowledged by the master. The frequency of those reports was low enough not to cause problems.

## Figures and Tables

**Figure 1 sensors-22-08708-f001:**
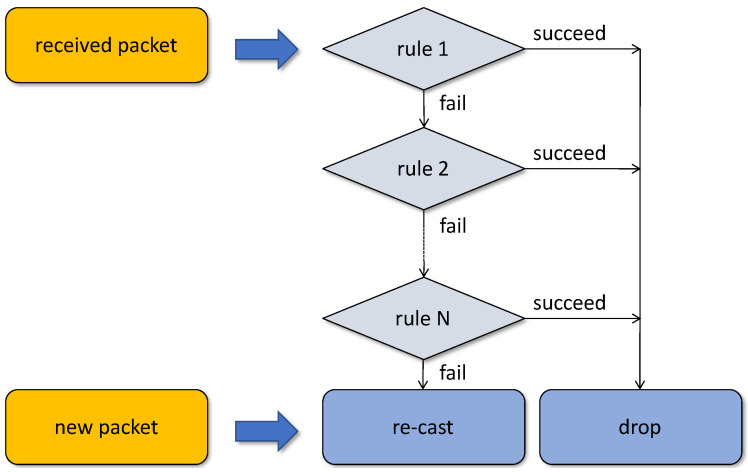
TARP rules.

**Figure 2 sensors-22-08708-f002:**
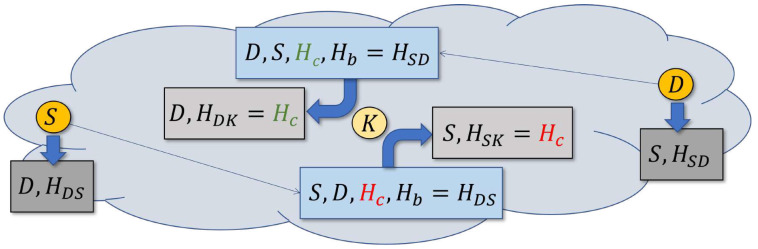
Data for the *SPD* rule.

**Figure 3 sensors-22-08708-f003:**
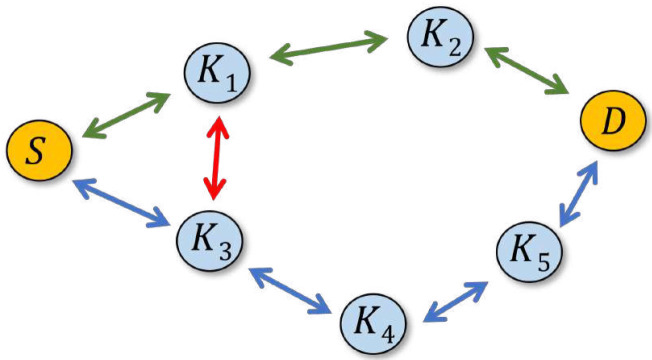
Fuzzy paths provided by the *SPD* rule.

**Figure 4 sensors-22-08708-f004:**
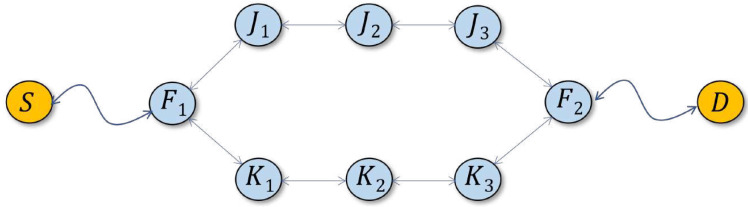
Multiple shortest paths.

**Figure 5 sensors-22-08708-f005:**
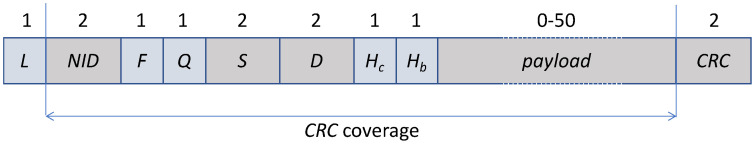
Base packet format in unsecured TARP.

**Figure 6 sensors-22-08708-f006:**
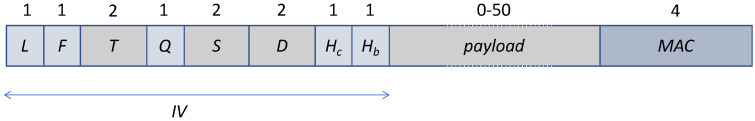
Packet format for secure TARP.

**Figure 7 sensors-22-08708-f007:**
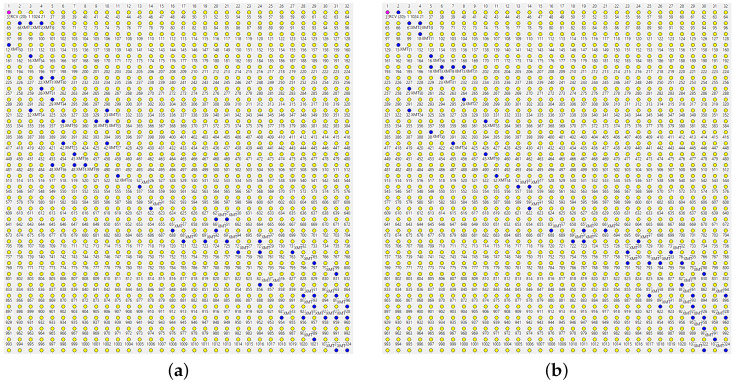
Two paths across the network. (**a**) Path 1; (**b**) path 2.

**Figure 8 sensors-22-08708-f008:**
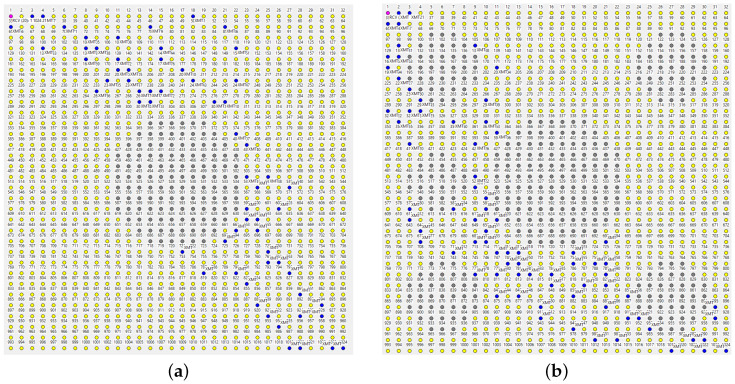
Forwarding around holes (moderate attack). (**a**) One hole; (**b**) six holes.

**Figure 9 sensors-22-08708-f009:**
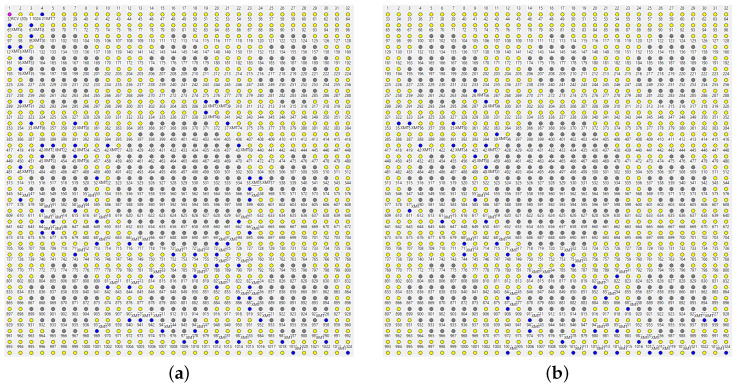
Forwarding around holes (massive attack). (**a**) Successful path; (**b**) failed path.

**Figure 10 sensors-22-08708-f010:**
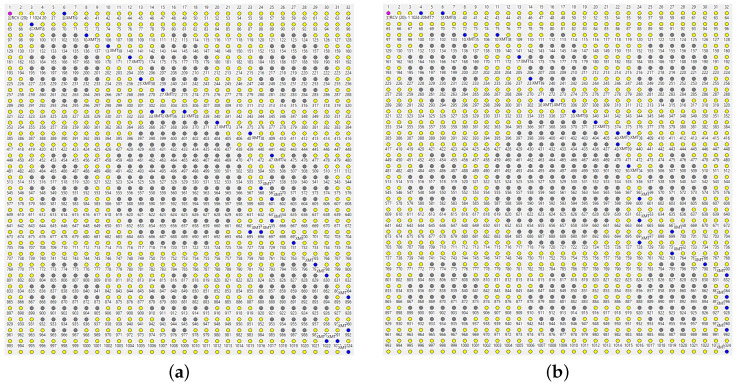
Two consecutive paths around nine holes following a fresh beacon. (**a**) Path one; (**b**) path two.

**Figure 11 sensors-22-08708-f011:**
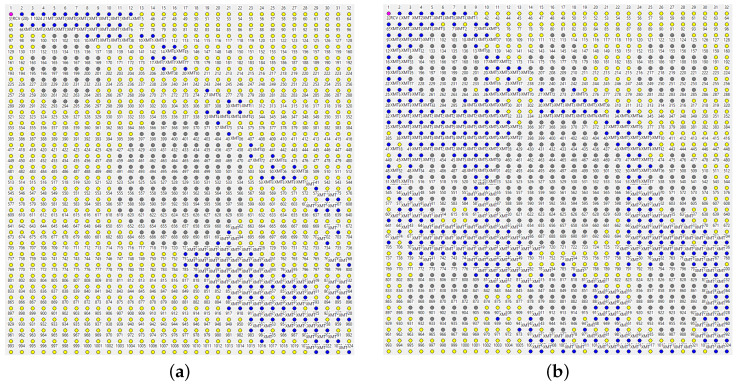
Paths obtained with A=1 and fuzzy ACKs turned on ($R_{a}=2$). (**a**) Under old beacon; (**b**) following a new beacon.

**Figure 12 sensors-22-08708-f012:**
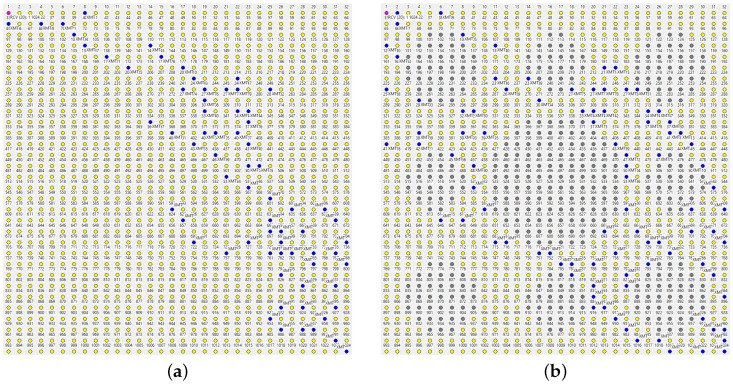
Paths for C=2 and A=0 (both within the same beacon interval). (**a**) Healthy network; (**b**) nine holes.

**Figure 13 sensors-22-08708-f013:**
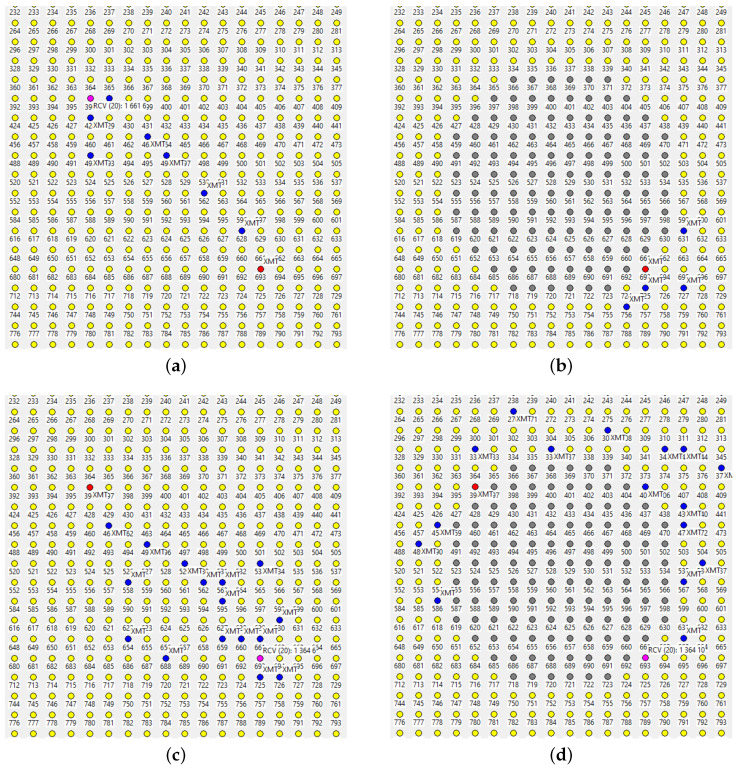
An RPC scenario with local and global relaxation options, C=2. (**a**) Local variant, no jamming; (**b**) local variant, jamming; (**c**) global variant, no jamming; (**d**) global variant, jamming.

**Table 1 sensors-22-08708-t001:** Current drain by CC1350.

Mode	*I*
Shutdown: device will wake up and reset on the nearest signal change on one of the preconfigured pins	185 nA
Standby: low-power idle state, basic clocks running, interrupts enabled, CPU halted waiting for an interrupt	1 μA
Idle: transient idle state with maximum-speed transition to Active	570 μA
Active: CPU running	2 mA
Radio RX: waiting for reception or receiving a packet	6 mA
Radio TX: transmitting a packet	23 mA

**Table 2 sensors-22-08708-t002:** Packet delivery fraction (PDF) on a single hop measured against light background load.

Distance (m)	PDF (%)
40.0	99.8
56.4	99.3
80.0	98.4
89.4	89.3
112.8	83.2
120.0	77.1
126.5	65.1

## Data Availability

The software (including the node firmware, the emulator, and the network models) used in this research is freely available (under GNU license) from Olsonet Communications at http://www.olsonet.com/software.

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
