# Peer review of "Rule-Driven Forwarding for Resilient WSN Infrastructures"

_sensors, 2022, doi:10.3390/s22228708_

Round 1

Reviewer 1 Report

The authors present TARP (Tiny Ad-hoc Routing Protocol) an extension of their previous works:
TARP: A Tiny Ad-hoc Routing Protocol for Wireless Networks (2003), A Tiny and Efficient Wireless Ad-hoc Protocol for Low-cost Sensor Networks (2007).

The paper is well written, in a good sequence.

TARP is an improvement over the classic Flooding protocol, in which some rules are added to reduce the amount of repeated messages forwarded by nodes.
Flooding is a stateless protocol, which does not need any network information to deliver a message from a source node to a destination node.
The rules LHC, RCV and DD, are normally included in any Flooding enhancements.

The SPD and SPP rules seem to be the contribution of TARP, already presented in both previous works.
SPD rule pushes TARP in AODV direction, all nodes will need routing tables, with the number of hops in destination direction. Routes are built opportunistically, there are no special packages to build them.
These tables will be very large if the network has many possible destinations, as in AODV applications, not just one sink node.
Stateless routing mechanisms for single-sink networks have been employed in recent works, as examples:
Fei Tong, Minming Ni, Lei Shu, Jianping Pan; A Pipelined-Forwarding, Routing-Integrated and Effectively-Identifying MAC for Large-Scale WSN; Globecom 2013
Ripudaman Singh, Biplab Sikdar; A Low Delay Routing-Integrated MAC Protocol for Wireless Sensor Networks; IEEE Internet of Things Journal, 2022;
Besides that GPSR (Greedy Perimeter Stateless Routing) is a classic protocol.

The rule SPP is not accurate, as discussed in line 378 "For SPP to work, the “parallel” nodes must be within mutual range"
In Flooding, many copies of messages are sent in many directions, drastically reducing SPP rule gains.

The contribution of this work seems to be the addition of Security and Resilience tasks.

Experimental results should be presented in a separate section, not a subsection on Security and Resilience.
The results presented in Fig. 7 seem to be obtained after a long sequence of messages sent from all nodes to node 1, as they probably obtained the H information. In the other case, the SPD rule would be useless, and the messages would be scattered throughout the network.

Readers would be more confident in the advantages of the TARP if the results were compared with other state-of-the-art protocols.

There are broken references on lines 864 and 866.

Reviewer 2 Report

The paper presents an interesting scheme for a WSN infrastructure. 

Some minor remarks:

1) in the energy budget of a node it is not included the energy cost of the sensors: are they always active? or are they periodically questioned ? 

2) each node holds only 1 sensor, is it right? if not, how to distinguish different inputs?

3) if the WSN nodes are fixed in space, the optimal path is already determined. the proposed scheme is useful, in my opinion, for moving nodes or under an attack

Reviewer 3 Report

The authors say that the paper improves the description and analysis of the TARP, which has been described elsewhere. They could and should better describe how the paper evolves over pre-existing papers on the same matter by the authors.

The paper is well-written and flows smoothly. Nevertheless, many discussions are non-causal. For example, Section 3 not only discusses “Relevant Issues” but also says what is not considered in TARP, assuming that TARP has been presented. At the same time, it was not, nor its basic strategy.

In line 157, the authors say that “the introduction of our forwarding scheme”; meanwhile, nothing has been said yet about the forwarding scheme.

The argument that Moore's Law is less relevant for WSN applications and deployments than for other fields seems to need an explanation.

Section 4 describes TARP. The links between the assumptions and observations: in 4.1 and the rules seem to be missing. TARP is broadcast-based, and how the rules can help to fulfill the assumptions and observations is welcomed. Besides, the broadcast mode seems to need the RF hardware to be on for larger periods than the alternatives, and a discussion on that seems to be missing.

Besides, while the authors place all tasks (rules and processing) in the application layer (owing to the so-called holistic viewpoint), many rules mimic link and network layer protocol features (efficiently implemented in hardware) at the application level running in the node’s processor (general purpose). It is unclear how this strategy may impact the power issues presented by the authors in the preceding sections.

The rationales for some comments, like the one in line 258 - “The recommended ordering of the rules is: LHC, DD, RCV”, are missing. In this case, it is also worth asking why not presenting the rules in this order.

“A” is used as node identification and slack parameter. Using the same notation for nodes’ identities and parameters should be avoided. Similarly, the notation for some control and header parameters are the same, confusing the reader.

Regarding the experimental setup, I am not sure that assuming that the nodes are deployed in a grid fashion is appropriate, moreover, for tactical WSN. This needs better reasoning, and probably a stochastic approach is better suited.

Typos for quotation marks/inverted commas should be corrected.

Please rephrase the sentence in line 377. This is not a textbook.

Many figure references are wrong. 

Reviewer 4 Report

A fuzzy routing scheme for  WSNs from the viewpoint of security in applications typical of tactical missions is presented.  Although the work is nicely presented and has significant potential, the authors failed to compare it to any competing forwarding or routing algorithms. It is of utmost importance to show the reader the advantages /disadvantages of your scheme compared to others.

Round 2

Reviewer 1 Report

Almost all recommendations were considered.
Comparisons of TARP with other routing protocols such as AODV and DSR have already been presented in both previous works by the authors, published in ATNAC03 and DATE07.
Authors are recommended to write some comments on the performance comparison made previously and justify the reason for using this protocol, highlighting the advantages of TARP.

Author Response

We have added text that appears as the third and fourth paragraph of the conclusions section which we hope presents the challenges of comparison with other schemes, it reads:

"
The paper does not discuss general performance issues related to TARP, focusing on the aspects related to security and resilience. A meaningful general quantitative performance comparison to other schemes, would have to account for many parameters including the distribution of nodes, channel characteristics, application demands (types of nodes, messages and their required delivery rates), and the parameters of TARP (which can be tweaked in several ways to tune the scheme to those demands), as well as those of the other schemes being compared. Additionally, such a comparison would still be open to criticism regarding its fairness due to factors such as TARP's lack of separation between ``data'' and ``control'' messages, or TARP's flexible approach to reliability which are not necessarily present or acceptable in other schemes. Consequently, none of our previous published work related to [2,5,14] compares its performance with that of other forwarding/routing schemes. 

The discussion in Section 6.1 provides powerful arguments in favor of TARP in comparison to the wider family of forwarding scheme that depend on point-to-point forwarding. A scenario where TARP could be be deficient to such schemes is one where traffic patterns with hard delivery time constraints (where the deterministic nature of fixed paths would become advantageous), or/and long data packets (where the data-link handshakes on the point-to-point hops would win over the simple LBT mechanism of TARP). Neither of these scenarios seem to be prevailing examples for typical WSN applications. TARP exemplifies a true adherent to the cross-layer networking paradigm. The real-life application of TARP presented in [5] provides a location-tracking functionality and is a revealing example which illustrates how the advantages of our scheme are fully realized via its holistic integration with the application. 

"

Reviewer 3 Report

The Paper has improved. 

I still have some concerns about the alleged stochasticity of the internal operation of the network's transmission/forwarding scheme. since the argument is reasonable, better leave it for the community to evaluate the claims.

I compliment the authors for their work. 

Author Response

We thank the reviewer for the support to our work.

Reviewer 4 Report

Authors has shown reasonable modifications given the complexity of the time given. I recommend publishing the manuscript with the modified version.

Author Response

(The authors gave the same response as above.)
